# Nanopore Detection Assisted DNA Information Processing

**DOI:** 10.3390/nano12183135

**Published:** 2022-09-09

**Authors:** Zichen Song, Yuan Liang, Jing Yang

**Affiliations:** 1School of Control and Computer Engineering, North China Electric Power University, Beijing 102206, China; 2Department of Computer Science and Technology, School of Electronics Engineering and Computer Science, Peking University, Beijing 100871, China

**Keywords:** nanopore detection, DNA storage, ONT nanopores, artificial intelligence, DNA information processing

## Abstract

The deoxyribonucleotide (DNA) molecule is a stable carrier for large amounts of genetic information and provides an ideal storage medium for next-generation information processing technologies. Technologies that process DNA information, representing a cross-disciplinary integration of biology and computer techniques, have become attractive substitutes for technologies that process electronic information alone. The detailed applications of DNA technologies can be divided into three components: storage, computing, and self-assembly. The quality of DNA information processing relies on the accuracy of DNA reading. Nanopore detection allows researchers to accurately sequence nucleotides and is thus widely used to read DNA. In this paper, we introduce the principles and development history of nanopore detection and conduct a systematic review of recent developments and specific applications in DNA information processing involving nanopore detection and nanopore-based storage. We also discuss the potential of artificial intelligence in nanopore detection and DNA information processing. This work not only provides new avenues for future nanopore detection development, but also offers a foundation for the construction of more advanced DNA information processing technologies.

## 1. Introduction

A storage system for electronic information is a fundamental component of modern information technology. However, with the advent of the era of big data, the storage capacity typically becomes inadequate for the system requirements. The surge in electronic information [1] has led to the development of long-term storage systems for big data that are based on magnets and semiconductors [2]. These systems come with unsustainable disadvantages, including limited lifespans [3], high infrastructure costs, and huge power consumption [4]. As a biological storage carrier of genetic information, DNA is an ideal medium for a next-generation storage system, offering the following advantages: ultra-high information density (455 EB data per gram [5]); long lifespan (half-life > 500 years [6,7]); relatively low energy consumption; programmability; and addressability [8]. Therefore, technologies that process DNA information have benefited from the cross-disciplinary integration between biology and information technology [9]. The three basic components of such technologies—DNA computation, DNA storage, and DNA self-assembly—provide a novel approach for the processing of large amounts of information.

The decoding of molecular information is essential to DNA information processing, where DNA storage is a typical example. In DNA storage, digital information can be encoded into DNA sequences according to specific algorithms, then stored in DNA strands by nucleotide synthesis technology and read out using DNA sequencing methods. The quality of DNA storage depends on DNA synthesis and sequencing [10,11]. Large-scale DNA can be synthesized by rapid low-cost solid-phase synthesis [12,13]. Traditional methods of DNA sequencing, such as Sanger sequencing [14] and Illumina sequencing, are costly and can result in the failure of large-scale DNA storage. Therefore, a highly efficient method for DNA sequencing is necessary to support the expansion of commercial applications of DNA information processing.

Nanopore detection is a single-molecule detection technique, and is useful for single-molecule chemistry studies [15], peptide and protein folding investigation [16,17], analysis of the mechanical aspects related to unzipping of nucleic acids or nucleic acids−protein complexes [18,19,20]. In the field of DNA information processing, nanopore detection is an efficient detection technique with two main applications: DNA nanopore sequencing and single-molecule sensing [21]. Nanopore sequencing technology with single nucleotide resolution is used to read information from DNA strands, thus assisting in error-free data recovery of large-scale DNA storage systems [22,23]. Additionally, nanopores can be assembled that act as single-molecule sensors for molecular identification. By modifying biological nanopores, the specificity and sensitivity of nanopores can be improved, thus expanding the detection range of molecules. Currently, a wide variety of molecules can be detected by nanopores, including DNA [24,25,26,27], DNA and RNA with lesions and nucleotide modification [28,29,30,31,32,33], whole cell nucleic acid preparation [34,35,36,37,38], unfolded protein peptides [39,40,41,42,43,44,45,46], and bacterial toxins [47]. Because of the efficient molecular identification capabilities of nanopores, DNA databases have more options for information storage. Take DNA modification as a typical example. DNA modifications are biochemically processed to bind DNA strands to other molecules. For the same DNA modification, the presence or absence of a modified nucleotide can be considered as a digital bit “1” and “0” along the oligonucleotide strand. Therefore, DNA modification sequences can be logically regarded as binary sequences. On the other hand, nanopores can precisely identify and distinguish multiple DNA modifications, which expands the alphabet of DNA storage and directly enhances potential storage density of DNA storage. At present, nanopore detection technique has been widely used in DNA storage systems.

This review was intended to cover recent advancements in nanopore detection technology and its application in DNA information processing. Therefore, we conducted a systematic investigation of the following: (1) principles and development history of nanopore detection technology; (2) developments in nanopore detection; (3) the two types of DNA storage that use nanopore detection; and (4) applications of artificial intelligence (AI) in nanopore data processing and DNA information technology. We envision this review article to advance the development of nanopore detection in the field of DNA information processing.

## 2. Principles and History of Single-Molecule Detection

Nanopore detection is a single-molecule detection technique that originated from patch clamp technology in 1976 [48]. Single-molecule detection features real-time monitoring and can be used to obtain the function, structure, dynamics, and other information of molecules [49]. According to the principles of detection, there are three methodological categories of single-molecule technology: optical, mechanical, and electrochemistry. Optical methods add fluorescent biological labels to molecules and observe the excitation or quenching of fluorescence through microscopy [50]. However, these methods change the structure of molecules, which can lead to inaccurate observations. Mechanical methods use atomic force microscopy [51] or scanning tunneling microscopy [52] to photograph biological molecular structures, and can provide direct images of biological molecules without fluorescent labels. Unfortunately, mechanical methods have several disadvantages that make them unsuitable for large-scale molecular detection: namely, bulky observation equipment, high cost and complex operations. Electrochemical-based single-molecule detection techniques generally rely on sensitive current monitors to detect molecules by analyzing ionic current signals [53]; the nanopore detection technique is one of its representatives.

### 2.1. General Principles of Biological Nanopore Detection Technology

In recent years, nanopore detection, whose advantageous features include no fluorescent-label requirement, portable equipment (Minion is only 90 g [54]) and low detection cost [55], has attracted much attention from researchers. The working principle of the experiment instrument for nanopore detection is similar to that of a Coulter counter [50], a single charged molecule is driven through the tiny pore embedded in a membrane by an electric field force, generating a transient current blocking signal. The general process of biological nanopore detection [56,57] begins by embedding a biological nanopore in an electrically resistant polymer membrane. A tank filled with an electrolyte solution is separated into two chambers by the membrane, and the nanopore becomes the only channel between the two chambers. Two electrodes are added to the chambers, and a constant voltage is applied to the electrodes; the negative voltage side is defined as “cis”, and the opposite side is defined as “trans”. During the detection process, the molecule of interest is driven along the nanopore under the influence of the applied electric field or/and the ensuing electroosmotic flow manifested inside ion-selective nanopores [58]. Charged biomolecules may be driven through the nanopores from “cis” to “trans” or in the opposite direction. Because of the nanoscale size internal diameter of the nanopore, biomolecules passing through the nanopore will hinder the flow of ionic current, producing discrete signal changes. Analysis of the signal characteristics of the ionic current through the use of a computer algorithm yields a variety of direct information about the target biomolecule, such as its species, structure and function.

### 2.2. Development of Nanopore Detection Technology

In the development process of nanopore detection, sequencing of DNA or RNA was initially focused on achieving high-quality base calling. The concept of DNA nanopore sequencing specifically originated in the late 1980s. In 1989, Hagan Bayley’s team began exploring the structure and function of oligomeric transmembrane protein pores such as α-hemolysin [59], and they later hypothesized that channel proteins could act as biosensors for molecules [60]. In 1996, Kasianowicz et al. was the first group to capture ionic current signals generated from single-stranded DNA (ssDNA) translocation through α-hemolysin nanopores [61], and, in 1999, Akeson et al. utilized α-hemolysin as a biosensor to achieve rapid discrimination between pyrimidine and purine segments along an RNA molecule [62]. More than a decade later, Gundlach et al. used *Mycobacterium smegmatis* porin A (MspA) [63] as a biological nanopore in combination with bacteriophage phi29 DNA polymerase to increase the number of identifiable DNA bases to ~30 [64]. Subsequently, in 2014, Oxford Nanopore Technologies (ONT) launched the first commercial nanopore sequencer, MinION, which has the advantages of single-molecule detection, long read length, fast sequencing speed, and portability [54].

### 2.3. Categories of Nanopores

Currently, there are two categories of nanopores based on raw material composition: protein nanopores and solid nanopores (made from solid materials). Due to their relatively small internal diameters, biological nanopores have a high signal-to-noise ratio and resolution. Moreover, with the advent of site-directed protein modification, biological nanopores can be used to detect a wide range of molecules, including DNA [65], RNA [66], protein, and metallic ions [67]. Biological nanopores include α-hemolysin (α-HL), MspA, *Escherichia coli* cytolysin (ClyA) [68], and aerolysin [69]. However, the structural stability of protein nanopores is easily affected by environmental conditions.

Compared to the biological nanopores, solid-state nanopores possess the advantages of excellent geometry flexibility, mechanical and chemical stability as well as compatible properties with modern semiconductor and microfluidics fabrication techniques [70,71,72,73,74,75]. Materials used to prepare solid-state nanopores include inorganic silicon [76], glass capillaries [77], and graphene [78]. Among these, silicon nitride is the most widely used material because of its low mechanical stress and excellent chemical stability. Although the excellent properties of solid-state nanopores hold great potential in biomolecular detection at single molecule level, they are still plagued by a number of particular drawbacks including high cost [79], poor reproducibility among different nanopores, large inner diameter, and lack of atomic-resolution functionalization [80].

## 3. Nanopores Detect Specific Modifications Carried by DNA Molecules

Nanopores can be used to simultaneously detect information about multiple changes or damage to single/double-stranded DNA that occurs during protein binding or modification by inorganic chemicals [81,82].

### 3.1. Detection of DNA Lesions

The accumulation of unrepaired DNA lesions may lead to premature cellular senescence, cancer, and some neurodegenerative diseases [83,84]. Currently, nanopores are being used as biosensors for DNA lesion detection. In 2019, Ma et al. proposed a nanopore detection method to identify cisplatin lesions on DNA [85] (Figure 1a). Cisplatin binds to *N*^7^ atoms on purines to form lesions on DNA, inhibiting the normal replication and transcription of DNA in cancer cells. Ma et al. applied MspA for the accurate detection of cisplatin-induced DNA lesions, demonstrating that the nanopore sequencing technique could identify cisplatin lesions in an input sequencing library of less than 10 ng. Furthermore, discrimination of multiple DNA lesions was achieved by observing the speed of DNA translocation through the nanopores. In 2022, Zhang et al. achieved direct identification of O6-carboxymethylguanine (O6-CMG), O6-methylguanine (O6-MG), and base-free (AP) sites through observation of the kinetics of enzymatic deceleration [86] (Figure 1b). They observed that progressive movement of phi29 DNA polymerase was hindered by DNA lesions such as O6-CMG, and recorded enzymatic arrest, suggesting that kinetic information generated by the interaction between a motor enzyme and DNA lesions can be used to identify multiple DNA lesions.

### 3.2. Detection of Nucleic Acids

The current nanopore sequencing techniques enable the detection of heterogenic nucleic acid. Heterogenic nucleic acids, also known as xeno-nucleic acids (XNAs), are a class of nucleic acid molecules with a non-natural backbone or nucleobase [87,88,89]. In 2019, Yan et al. presented a method for direct sequencing of 2-deoxy-2-fluoroarabinoic acid (FANA) through nanopore-induced phase shift sequencing (NIPSS) [90] (Figure 1c). By ligating FANA with a DNA drive-strand, they showed that the direct sequencing of FANA could be achieved by NIPSS with phi29 DNA polymerase. Their contribution led to the development of a universal identification method based on nanopore sequencing that can be used to clearly distinguish between DNA, RNA and XNA nucleotides.

### 3.3. Detection of Peptides and Proteins

Protein detection can also be achieved using nanopore sequencing techniques [20,91,92]. In 2021, Wang et al. observed single molecule ratcheting of peptides with MspA [93] (Figure 1d). By constructing peptide−oligonucleotide conjugates (POCs) and using NIPSS for measurements, they directly observed the discrete steps of the ratcheting motion of a peptide. Experimentally, the current signal results of peptides generated from NIPSS measurements show a highly consistent pattern, with a clear correlation to the amino acid sequence of the peptide. Subsequently, in November 2021, Brinkerhoff et al. demonstrated a nanopore-based peptide sequencing method with single-amino acid resolution [94]. They designed an experiment in which a POC was driven through an MspA nanopore by a DNA helicase, and they were able to observe steplike current signals generated by a peptide ratcheting through the nanopore. Additionally, they changed amino acids at fixed sites on the peptide sequence and observed the resulting changes in the ionic current sequence, enabling protein sequencing.

### 3.4. Detection of Inorganic Chemical Molecules

In addition to detecting biomolecules, nanopores can be used to detect inorganic chemical molecules. In 2021, Jia et al. proposed a programmable nanoreactor for random stochastic sensing (PNRSS) based on a nanopore sequencing technique, enabling real-time monitoring of chemical reactions at the single-molecule level [95] (Figure 1e). A wide range of single-molecule reactions of metal ions, simple organic compounds such as lactic acid, and nucleoside analogues have been directly observed through PNRSS. Moreover, they used AI tools to enhance the sensing resolution of PNRSS, which ultimately allowed them to detect a total of 20 types of chemical reactions.

**Figure 1 nanomaterials-12-03135-f001:**
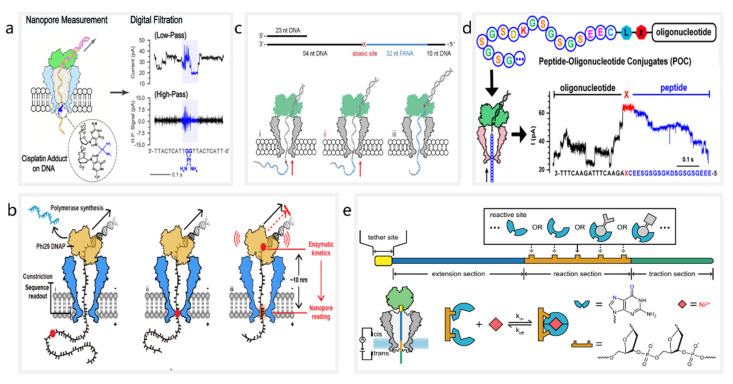
The schematic diagrams of nanopore-based molecular detection systems. (**a**) Detection of cisplatin lesions on DNA. Reproduced with permission from Fubo Ma, ACS Sensors; published by American Chemical Society, 2021. (**b**) Stalling kinetics readout during nanopore sequencing using an MspA nanopore (blue) and a phi29 DNAP (yellow). Reproduced with permission from Jinyue Zhang, Nano Letters; published by American Chemical Society, 2022. (**c**) Sequencing of chimeric DNA (grey) -FANA (cyan) with an abasic spacer (red). Reprinted from [90]. (**d**) Ratcheting motion of a POC using NIPSS. Reproduced with permission from Shuanghong Yan, Nano Letters; American Chemical Society, 2021. (**e**) Detection of inorganic chemical molecules through PNRSS. Reprinted from [95].

## 4. DNA Storage Based on Nanopore Sequencing Technology

Although the nucleotide strands carrying digital information are fragile, which may be interfered with and destroyed by external environmental factors during the preservation, such as ultraviolet rays, extreme temperature changes, and biological contamination such as bacteria and viruses. However, with properties of ultra-high information density and long lifespan, DNA is expected to become a novel data storage medium for the next generation of information processing systems. We have reason to believe that with the advancement of DNA preservation techniques and preservation equipment [96,97,98,99], the capacity, duration and storage quality of DNA storage will be greatly improved. Currently, DNA nanopores can be used to identify sequences of DNA/RNA and molecular or chemical modifications of DNA/RNA, all of which can be considered as bit sites, thereby providing more options for DNA storage systems. There are two categories of DNA as a carrier for digital data storage: (1) synthetic DNA base sequences; and (2) DNA nanostructures/modifications.

### 4.1. DNA Storage Based on Synthetic DNA Sequences

As illustrated in Figure 2a, the entire procedure of DNA storage is commonly divided into six steps [100] (Figure 2a): (1) encoding digital information into DNA sequences; (2) designing and synthesizing DNA sequences; (3) preserving the DNA in vivo or in vitro; (4) random access of specific DNA sequences; (5) reading of the specific DNA sequences; and (6) decoding and recovering DNA sequences into digital information. At present, nanopore sequencing technique is widely used in steps 4 and 5.

Nanopore sequencing technology is advantageous for the data decoding of DNA storage systems. In 2018, Lee et al. designed and validated a large primer library using over 13 million oligonucleotides stored in 35 files and totaling 200 MB of data, and they were able to achieve error-free data recovery of all DNA files using random access methods and nanopore sequencing [23]. In 2019, Lopez et al. demonstrated a method for DNA storage that combines random access, DNA assembly, and nanopore sequencing [101] (Figure 2b). They employed the MinION sequencer to successfully recover digital information stored in 111,499 oligonucleotides and totaling 1.67 terabytes of data. This method allows for an approximately 100-fold increase in sequencing and decoding capacity compared with previous reports using nanopores in a DNA storage system.

Nanopore sequencing technology can be applied to both in vivo and in vitro storage systems. In 2021, Chen et al. designed and synthesized an in vivo system using an artificial yeast chromosome of 254,886 bp [22] (Figure 2c). The chromosome was written into 37,782 bits of data using sparse low-density parity-check (LDPC) codes and pseudo-random sequences, comprising a total of two images and a video clip. During the DNA information reading stage, they used a nanopore sequencer to achieve accurate base calling, achieving reliable data recovery at an error rate of 10.79%.

### 4.2. DNA Storage Using DNA Nanostructures and Modifications as Information Carriers

Nanopore sequencing techniques that provide accurate identification on DNA modifications or nanostructures offer new solutions for DNA storage. Highly programmable DNA nanostructures offer a variety of address sites for storage of digital data [102,103]. In 2018, Chen and Kong et al. proposed a DNA storage scheme that considered DNA hairpins as bit sites [104] (Figure 3a). In their work, DNA hairpins of different lengths were regarded as digital bits and used to develop a high-resolution solid-state nanopore sequencing method. DNA hairpins of 8 bp and 16 bp in length could be clearly distinguished using quartz nanopores with an internal diameter of ~5 nm. Thus the 8-bp and 16-bp hairpins were assigned bit-0 and bit-1, respectively, and used to attach 56 hairpins to a 7228-bp long oligonucleotide, thereby forming a 56-bit storage segment. Using a similar idea, Bell and Keyser [105] designed a DNA nanostructure library based on the principles of DNA origami in which each member has a unique barcode, and each bit on a bar code is represented by the presence or absence of a DNA dumbbell hairpin. They eventually confirmed that a 3-bit barcode could be recognized with 94% accuracy through solid-state nanopore sequencing.

An alternative DNA storage system of nanopores is being used for identification of biopolymer sequences. In 2020, Cao et al. used specially tailored biopolymer sequences as bit information storage carriers [106] (Figure 3b). The biopolymer sequences are biohybrid macromolecules comprising two different-sized monomers (n-propyl phosphate and [2,2-diynyl]-propyl phosphate) and natural nucleotides, where the monomers are mapped as bit-0 and bit-1, respectively. The study used bioengineered nanopores of the aerolysin toxin to successfully achieve bit-site recognition of the customized biopolymers with single-base resolution. Additionally, deep learning was applied to achieve high-precision encoding and decoding of up to 4-bit digital sequences. This unique system provides inspiration for the development of new DNA storage systems.

## 5. Applications of AI in Nanopore Data Processing and DNA Information Technology

The integration of AI with DNA information processing has led to unexpectedly good outcomes in recent years. Deep learning is a branch of AI that mainly uses multi-layer neural networks to learn from data. Deep learning models can be used to automatically learn from and extract features of raw data, and they have powerful capabilities for mining the potential rules of big data. Models of deep learning such as convolutional neural networks (CNNs), deep confidence networks, and recurrent neural networks (RNNs) have been developed that perform well on multiple tasks, including computer vision, speech recognition, and natural language processing [107,108,109]. According to previous reports [110,111], the huge amount of sequence data generated from nanopore detection has led to the design and training of multiple deep learning models that are being widely used in DNA information processing tasks, such as base calling, biomolecular detection, and DNA storage.

### 5.1. Base Calling

Base calling is the process of inferring the order of nucleotides in a DNA segment during sequencing [112]. Because nanopore sequencing generates current signals, base calling requires computer algorithms to process the sequence data. To date, many researchers, including the team at ONT, have designed a variety of software programs based on deep learning models for base calling. These software programs can be categorized by the two types of input data: segmentation events and raw current signals.

Early base calling software relies on the analysis of segmented events. In 2016, David et al.’s research and development team developed Nanocall, the first open-source, offline basecaller for Oxford nanopore sequencing data [113]. It uses a hidden Markov-based model (HHM) for base sequence identification. Using the R7.3 version of the MinION, Nanocall analyzed data from two *Escherichia coli* and two human genetic samples and found reads with 68% identity. Because HHM is not suitable for long homogeneous polymer detection, RNNs are applied to segmented event sequences. In 2016, Boza et al. proposed DeepNano, an open-source DNA basecaller with deep RNNs [114]. Using R7.3 test datasets for *E. coli* and *Klebsiella pneumoniae* to evaluate the basecaller accuracy of DeepNano, they found that, for 2D reads, DeepNano achieved the base recognition accuracies of 88.5% and 86.7%, respectively. In terms of the speed of base calling, DeepNano is 5 to 20 times faster than Nanocall.

The direct conversion of raw current signals into base sequences by deep learning models is convenient and accurate. BasecRAWller, a base-calling software for nanopore data based on raw current signals, was proposed by Stoiber et al. in 2017 [115]. BasecRAWller uses a pair of unidirectional RNNs to make real-time DNA base calls directly from raw nanopore reads and has been evaluated on the basis of its performance with two data sets: *E. coli* and human. Reportedly, BasecRAWller reads have 81.7% and 72.5% identities on the *E. coli* and human datasets. Soon thereafter, Teng et al. reported on Chiron, the first deep learning model to implement end-to-end base calls and convert raw current signals directly into nucleotide sequences [116]. Chiron combines a CNN with an RNN and a connectionist temporal classification decoder, which allows it to learn directly from the raw signal data without using event segmentation. Chiron achieves 90.57% and 81.54% identities on *E. coli* and human datasets, respectively, which are higher than those achieved by the other three software programs mentioned above. In terms of speed on a central processing unit processor, Chiron is slower than BasecRAWller (21 bp/s vs. 81 bp/sec, respectively). Moreover, Chiron is fully open source, allowing users to train their own neural networks and develop specialized base-calling applications.

### 5.2. Biomolecule Detection

Deep learning is a powerful tool for improving the nanopore identification accuracy of multiplex biomolecules, which can be effectively used in the detection of biomolecular modifications. In 2019, Liu et al. designed and trained DeepMod [117] (Figure 4a), a bidirectional RNN model with short long-term memory (LSTM) that is suited for high-precision DNA modification detection from raw current signal extracted by the ONT nanopore sequencer. DeepMod was evaluated in the nanopore readouts of the *E. coli*, *Chlamydomonas* reinhardtii, and human genomes. The results showed that DeepMod can detect methylation-modified DNA nucleotides with high accuracy; for example, 5-methylcytosine (5 mC) and 6-methyladenine (6 mA) exhibited average detection accuracies of 0.99 and 0.9, respectively. Similarly, in 2021, Ni et al. designed a deep learning tool called DeepSignal-plant that delivers genome-wide detection of all three sequence contexts of cytosine methylations that are naturally occurring in plants using nanopore reads [118] (Figure 4b). With an architecture of multilayer bidirectional RNN followed by a full connection layer, DeepSignal-plant can automatically extract and learn both sequence features and signal features from nanopore data, and is one-eighth the size of its predecessor, DeepSignal [119].

### 5.3. DNA Storage

AI can be integrated into multiple steps of DNA storage, potentially accelerating the reading speed of oligonucleotides, providing efficient and accurate random-access methods, and promoting the further realization of commercialized large-scale DNA storage systems.

Deep learning holds great potential in the rapid analysis of nanopore reads. In 2020, Nivala et al. proposed a method for tagging physical objects using DNA or other molecules in situations where traditional methods such as radiofrequency identification tags and quick response codes do not apply [120] (Figure 4c). They developed the Porcupine system, an end-user molecular tagging system with the ability to read DNA-based tags in seconds using a portable nanopore device. Its digital bits are represented by the presence or absence of different DNA strands, called molecular bits (molbits), which are classified by a CNN directly from the raw nanopore signal. This method avoids the need for base calling using DNA sequences and thus greatly reduces the time requirement and complexity.

In summary, deep learning can be applied to rapidly increase the base recognition accuracy of DNA sequencing (identities increased from 68% to 90.57%), strongly expand the range of molecular types that can be detected by nanopores, and provide the possibility of high-speed DNA reading, all of which are crucial factors for the large-scale commercialization of DNA storage and nanopore technologies.

Deep learning can also be used to scale up the capacity of DNA storage systems with computer simulation, which provides helpful suggestions to guide research. Having access to a large-scale DNA storage system is equivalent to being able to design complex DNA primer sequences, which are unaffordable in most research settings. Therefore, precise control and prediction of the DNA hybridization process are critical for the design of large-scale DNA storage systems. In 2021, David Buterez was the first to present a comprehensive study of a machine learning technique for DNA hybridization prediction [121] (Figure 4d). As a baseline, he conducted performance evaluations of multiple machine learning models on an *in silico*-generated hybridization dataset containing more than 2.5 million DNA sequence pairs. Next, he evaluated this dataset using CNN, RNN and RoBERTa models and found that the deep learning models delivered more accurate DNA hybridization prediction and reduced the running time by one to two orders of magnitude compared with the baseline models.

## 6. Conclusions

DNA information processing technologies utilize the DNA molecule as a data storage medium and data calculation unit and have the potential to store big data. However, the development of these technologies has been hampered by the high cost, low speed, and relatively low accuracy of traditional DNA sequencing. Nanopore detection is a new single-molecule detection technique that enables molecular identification through the analysis of ionic current signals generated by molecules passing through nanopores. Compared with other methods, rapid nanopore detection offers the advantages of being label-free, low-cost, and convenient, and satisfies the requirements of DNA information processing. At present, nanopore detection is widely used in DNA information processing tasks, such as biomolecular detection and DNA storage. However, the development of nanopore detection is encountering obstacles that may hinder further improvements in DNA information processing.

Nanopores are frequently employed in biomolecular detection, due to improvements in specificity and sensitivity, and are useful in the detection of a variety of molecules, including nucleic acids, proteins, and inorganic ions. Nevertheless, a nanopore detection platform with excellent performance has not been systematically unified to eliminate repetitions of experiments in different labs. In fact, different biological nanopores have been independently designed by several research teams to detect molecules. Due to a lack of agreed standards, there is a range of reported detection results for the same molecules, potentially hindering the development of biomolecular detection.

In DNA storage systems, digital information can be stored in DNA sequences, sequences containing modified DNA, or other biomolecules. Nanopore technology can optimize the reading process of DNA information, providing high-speed, accurate base calling in the field of DNA storage. However, DNA nanopore sequencing remains limited in two aspects. First, although the base-calling accuracy of nanopore sequencing has improved greatly, currently ranging 90–95%, it is still lower than the 99% accuracy of next-generation sequencing [122]. Second, the throughput of nanopore detection is relatively low. Nanopore detection relies on microcurrents from molecules passing through nanopores, and simultaneous detection of multiple nanopores can lead to signal distortion due to superposition of electrical signals. The current read speed for a single nanopore is ~10 ms/base, which would take approximately 20 years to sequence the human genome at a depth of 10× coverage [123]. Therefore, higher sequencing accuracy and detection throughput could promote wider commercial applications of large-scale DNA storage.

The application of AI in nanopore technology can be expected to overcome the above obstacles. In biomolecular detection, the powerful function of pattern recognition within deep learning models can allow for simultaneous detection of a wide range of molecules. Additionally, AI offers high-precision sequence prediction, thereby improving the speed and accuracy of base calling for DNA storage. Finally, AI can also efficiently predict molecular folding of proteins, which is useful to modify structures of biological nanopores, potentially providing a new avenue to improve detection throughput. With the help of more advanced nanotechnologies and AI, we anticipate that DNA nanopores will continue to provide new applications in the area of DNA information processing.

## Figures and Tables

**Figure 2 nanomaterials-12-03135-f002:**
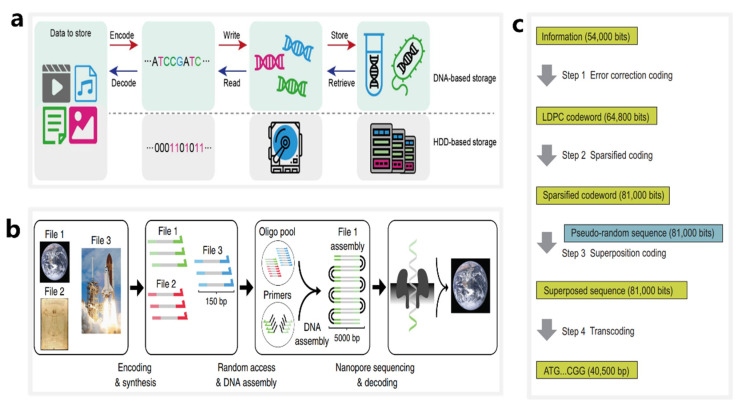
The workflows of DNA storage systems based on synthetic nucleotide sequences. (**a**) The major steps of digital data storage in DNA corresponding to conventional hard-disk storage. Reproduced with permission from Yaya Hao, SMALL STRUCTURES; published by John Wiley and Sons, 2020. (**b**) An overview of a DNA data storage workflow using ONT nanopores as tools to sequence long double-stranded DNA strands obtained by random access. Reprinted from [101]. (**c**) The workflow of an in vivo DNA storage system using an artificial yeast chromosome. Reprinted from [22].

**Figure 3 nanomaterials-12-03135-f003:**
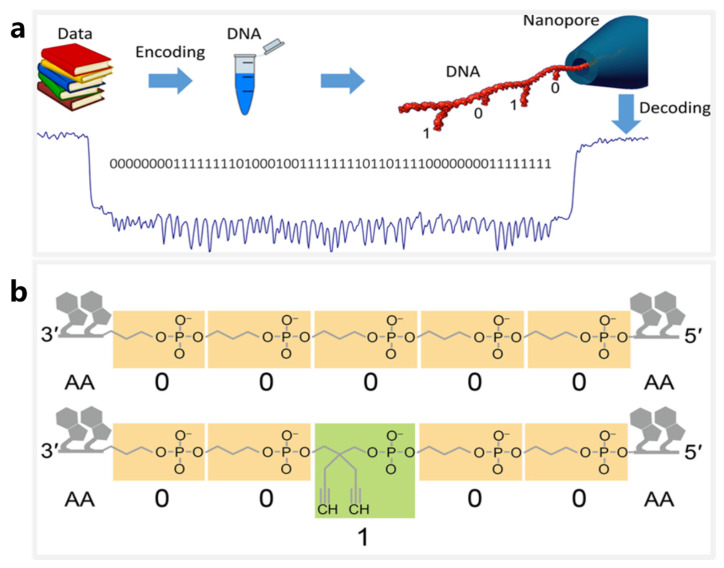
Molecular storage systems based on DNA nanostructures. (**a**) A schematic of the measurement of a DNA carrier using a nanopore, where bits ‘1’ and ‘0’ represent DNA hairpin structures of 16 bp and 8 bp, respectively. Reproduced with permission from Kaikai Chen, Nano Letters; American Chemical Society, 2019. (**b**) An illustration of biopolymer sequences, where ‘0’ represents a monomer molecule and ‘1’ represents its methylated version. Reprinted from [106].

**Figure 4 nanomaterials-12-03135-f004:**
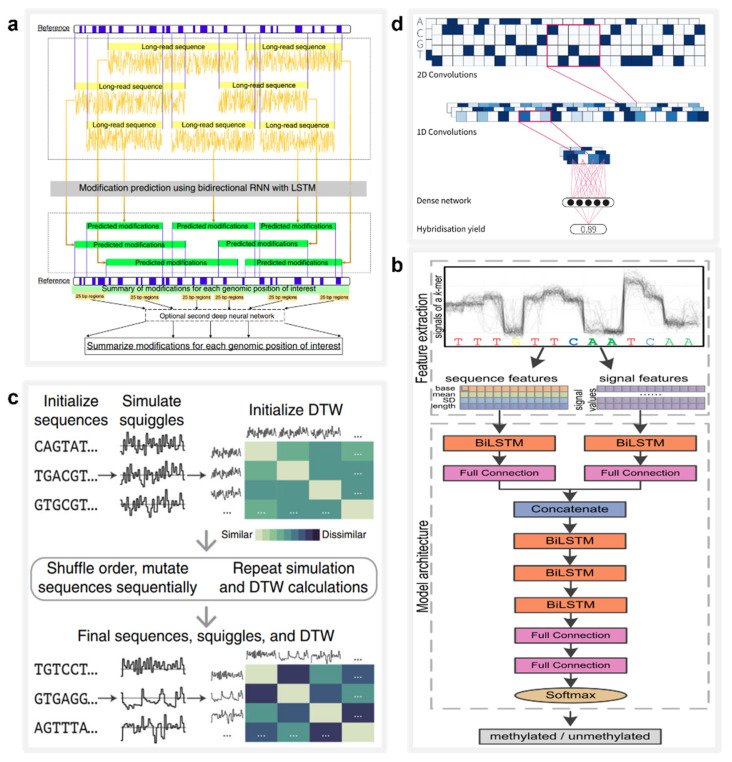
The applications of AI in DNA information storage. (**a**) Architecture of the neural network model DeepMod, which is used to capture the time-series characteristics of nanopore signals and detect DNA modifications. Reprinted from [117]. (**b**) Architecture of DeepSignal-plant, in which sequence and signal features can be extracted using bidirectional LSTM (biLSTM). Reprinted from [118]. (**c**) Evolutionary model workflow for hybrid prediction DNA coding. Reprinted from [120]. (**d**) Schematic diagram of CNNs used to quickly read molecular labels. Reprinted from [121].

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
