# Peer review of "Nanopore Detection Assisted DNA Information Processing"

_nanomaterials, 2022, doi:10.3390/nano12183135_

Round 1
Reviewer 1 Report
11. Authors state that: ‘Nanopore detection is a single-molecule detection technique with two main applications: DNA nanopore sequencing and single-molecule sensors’.
In my opinion this is an understatement; the rich pool of literature demonstrated that besides the aforementioned applications, the NP detection is useful for single-molecule chemistry studies (Cao, C.; Long, Y.-T. Biological Nanopores: Confined Spaces for Electrochemical Single-Molecule Analysis. Acc. Chem. Res. 2018, 51, 331−341; ), peptide and protein folding investigation (Oukhaled, A.; Bacri, L.; Pastoriza-Gallego, M.; Betton, J. M.; Pelta, J. Sensing Proteins through Nanopores: Fundamental to Applications. ACS Chem. Biol. 2012, 7, 1935−1949.; Mereuta, L.; Asandei, A.; Seo, C. H.; Park, Y.; Luchian, T. Quantitative Understanding of pH- and Salt-Mediated Conformational Folding of Histidine-Containing, β -Hairpin-Like Peptides, through Single-Molecule Probing with Protein Nanopores. ACS Appl. Mater. Interfaces 2014, 6, 13242−13256.), analysis of the mechanical aspects related to unzipping of nucleic acids or nucleic acids−protein complexes (Hornblower, B.; Coombs, A.; Whitaker, R. D.; Kolomeisky, A.; Picone, S. J.; Meller, A.; Akeson, M. Single-Molecule Analysis of DNAProtein Complexes Using Nanopores. Nat. Methods 2007, 4, 315−317.; Mathe, J.; Visram, H.; Viasnoff, V.; Rabin, Y.; Meller, A. Nanopore Unzipping of Individual DNA Hairpin Molecules. Biophys. J. 2004, 87, 3205−3212.; Luchian, T., Park, Y., Asandei, A., Schiopu, I., Mereuta, L., & Apetrei, A. (2019). Nanoscale probing of informational polymers with nanopores. Applications to amyloidogenic fragments, peptides, and DNA–PNA hybrids. Accounts of Chemical Research, 52, 267–276). Given the nature of the review, I suggest that authors amend their statement and possibly cite relevant literature, as mentioned above, for the sake of unraveling previous contributions in the field.
22. Authors state that ‘According to the principles of detection, there are two methodological categories of single-molecule technology: optical and mechanical.’. This is again an incomplete account of the facts, as authors dismiss probably the most relevant method, namely the electric recordings; however, a mention is made in this respect (‘Nanopore detection is a single-molecule detection technique that originated from patch clamp technology in 1976’), so authors contradict themselves. I suggest correction.
33. Authors state that ‘The principles of nanopore detection and the Kurt counter [50]..’..the authors meant probably the Coulter and not Kurt counter.
Moreover, in the continuation of the sentence, authors state that ‘..detect molecules by analyzing the ionic current signals generated from a tiny pore..’. This is not completely accurate, as the Coulter counted detects macroscopic objects, e.g cells, whereas only the nanopore-based sensors, due to their nanoscopic inner diameters, detect molecules and/or ions. I suggest correction.
44. Authors state that ‘During the detection process, a stable ionic current is formed by the directional movement of ions through nanopores under the force of an electric field’. This is again not entirely true, as electroosmotic forces were shown to greatly contribute to the molecular capture process, see: ‘Firnkes, M.; Pedone, D.; Knezevic, J.; Doblinger, M.; Rant, U.
Electrically Facilitated Translocations of Proteins through Silicon Nitride Nanopores: Conjoint and Competitive Action of Diffusion, Electrophoresis, and Electroosmosis. Nano Lett. 2010, 10, 2162−2167
I recommend proper text amendment and citation.
55. Authors state that ‘Solid-state nanopores have stable mechanical and chemical properties and are not easily affected by external environmental conditions, especially temperature. For the benefit of the reared, a broader discussion presenting also the existing drawbacks of solid-state nanopores (e.g., poor reproducibility among different nanopores, lack of atomic-resolution functionalization) should be inserted.
66. Authors state that ‘Protein detection can also be achieved using nanopore sequencing techniques.’; however, proper citation of relevant work in the field is missing. Again, given the nature of the review and for the benefit of the reader, a number of relevant papers should be also cited, including:
Piguet, F.; Ouldali, H.; Pastoriza-Gallego, M.; Manivet, P.; Pelta, J.; Oukhaled, A. Identification of Single Amino Acid Differences in Uniformly Charged Homopolymeric Peptides with Aerolysin Nanopore. Nat. Commun. 2018, 9, 966
Luchian, T., Park, Y., Asandei, A., Schiopu, I., Mereuta, L., & Apetrei, A. (2019). Nanoscale probing of informational polymers with nanopores. Applications to amyloidogenic fragments, peptides, and DNA–PNA hybrids. Accounts of Chemical Research, 52, 267–276
Movileanu, L.; Schmittschmitt, J. P.; Scholtz, J. M.; Bayley, H. Interactions of Peptides with a Protein Pore. Biophys. J. 2005, 89, 1030−1045.
H. Ouldali, K. Sarthak, T. Ensslen, F. Piguet, P. Manivet, J. Pelta, J. C. Behrends, A. Aksimentiev, A. Oukhaled, Nat. Biotechnol. 2019, https://doi.org/10.1038/s41587-019-0345-2.
Reviewer 2 Report
The review entitled "Nanopore Detection Assisted DNA Information Processing" publishes the results of research on the detection of nanopores and the processing of DNA information. The review is logical, written in understandable language and is of scientific interest. The number of literary sources is sufficient, references to them are appropriate. It is proposed to accept the article after minor changes.
1) In the first paragraph of the introduction, it would make sense to add a short message about the successful use of magnetic composite materials with DNA by researchers, for example, https://doi.org/10.3390/polym14020344 .
2) The introduction says "For the same DNA modification, the presence or absence of modified nucleotides can be considered as digital bits "1" and "0"." But there may be more than one modification of nucleotides, which will complicate the transition to binary code.
3) Figure 2 is slightly shifted to the left.
4) Chapter 4. "It is expected that due to its ultra-high information density and long service life, DNA will become a new means of data storage for the next generation of information processing systems." As a counterargument, we can note the fragility of the DNA molecule and its susceptibility to destruction under the influence of a number of factors, for example, ultraviolet.
5) Chapter 5. "deep learning can be applied to rapidly reduce the error rate of DNA sequencing" Please clarify how?
6) Chapter 6. "Nevertheless, a nanopore detection platform with excellent performance has not been successfully commercialized." Here, rather, it is not about commercialization, but about the unification of the system to eliminate repetitions.
Round 2
Reviewer 1 Report
The revised version of the manuscript looks much improved, although certain minor points need to be addressed before full acceptance.
1. For instance, authors state that:
'.. Although the excellent properties of solid-state nanopores hold great potential in biomolecular detection, solid-state nanopores have been unable to achieve single molecule sensing..'
This is grossly inaccurate; despite the fact that SSN have the drawbacks described by the authors thereafter, they (SSN) are still able to achieve single-molecule sensing, there are plenty of good references in this respect, some of them were also cited in this manuscript.
I'd suggest a modification of that paragraph, as follows:
' Although the excellent properties of solid-state nanopores hold great potential in biomolecular detection at single molecule level, they are still plagued by a number of particular drawbacks including...' (and here follow the drawbacks mentioned by the authors themselves)
2. Another issue which needs addressing relates to English used; for instance, the sentence:
'During the detection process, a stable ionic current is formed by the directional movement of ions through nanopores under the force of an electric field and the force of electroosmotic..' needs changing.
As a suggestion, I recommend:
'During the detection process, the molecule of interest is driven along the nanopore under the influence of the applied electric field or/and the ensuing electroosmotic flow manifested inside ion-selective nanopores..'
